# Role of the Laparoscopic Approach for Complex Urologic Surgery in the Era of Robotics

**DOI:** 10.3390/jcm10091812

**Published:** 2021-04-21

**Authors:** Iulia Andras, Angelo Territo, Teodora Telecan, Paul Medan, Ion Perciuleac, Alexandru Berindean, Dan V. Stanca, Maximilian Buzoianu, Ioan Coman, Nicolae Crisan

**Affiliations:** 1Department of Urology, Iuliu Hatieganu University of Medicine and Pharmacy, 400012 Cluj-Napoca, Romania; dr.iuliaandras@gmail.com (I.A.); vasilestanca@yahoo.com (D.V.S.); maximilian.buzoianu@yahoo.com (M.B.); jcoman@yahoo.com (I.C.); drnicolaecrisan@gmail.com (N.C.); 2Department of Urology, Municipal Hospital, 400139 Cluj-Napoca, Romania; medan.paul@gmail.com (P.M.); ion.perciuleac@yahoo.com (I.P.); dr.berindean@gmail.com (A.B.); 3Uro-Oncology and Kidney Transplantation Unit, Department of Urology, Fundacio Puigvert, Autonoma University of Barcelona, 08025 Barcelona, Spain; territoangelo86@gmail.com

**Keywords:** 3D laparoscopy, dual combined approach, radical cystectomy, radical nephroureterectomy, robotic surgery

## Abstract

(1) Introduction: The advent of robotic surgery led to the assumption that laparoscopic surgery would be replaced entirely. However, the high costs of robotic surgery limit its availability. The aim of the current study was to assess the feasibility of the 3D laparoscopic approach for the most complex urological procedures. (2) Materials and methods: We included in the current study all patients who had undergone complex 3D laparoscopic procedures in our department since January 2017, including radical nephrectomy (LRN) using a dual combined approach (19 patients), radical nephroureterectomy (LRNU) with bladder cuff excision (13 patients), and radical cystectomy (LRC) with intracorporeal urinary diversion (ICUD) (21 patients). (3) Results: The mean operative time was 345/230/478 min, the complications rate was 26%/30.76%/23.8% and positive surgical margins were encountered in 3/1/1 patients for the combined approach of LRN/LRNU/LRC with ICUD, respectively. A single patient was converted to open surgery during LRN due to extension of the vena cava thrombus above the hepatic veins. After LRC, sepsis was the most common complication and 8 patients were readmitted at a mean of 15.5 days after discharge. (4) Conclusions: In the era of robotic surgery, laparoscopy remains a plausible alternative for most complex oncological cases.

## 1. Introduction

Minimally invasive surgery has delivered great benefits for patients, improving perioperative outcomes such as estimated blood loss, postoperative pain, and wound infection and reducing the average hospital stay as compared with that following open surgery [1]. Studies assessing the quality of life (QoL) after surgery have shown that, in comparison to open procedures, minimally invasive procedures achieve better mental QoL 2 days after the intervention and better physical QoL at 1 month [2]. Furthermore, similar oncologic and functional postoperative outcomes have been reported for several surgical procedures when comparing open and minimally invasive approaches [3]. 

While minimally invasive surgery was initially reserved for selected cases, the introduction of robotic surgical systems has allowed its use in more complex and challenging procedures [4]. By the end of 2017, 5770 platforms had been shipped worldwide, the total revenue being estimated at US $3.1 billion dollars, and more than 877,000 robotic procedures were performed globally in that year, mainly in the fields of general surgery, gynecology, and urology [5]. Another study found that robotic surgery was used 8.4 times more frequently in 2018 compared with 2012 [6]. 

At first it was thought that laparoscopic surgery would ultimately be replaced entirely by the robotic approach, but the development of improved laparoscopic systems opened up new possibilities [7]. Three-dimensional (3D) laparoscopy has gained increasing popularity, providing quality in-depth vision of the work space and reducing the overall task performance time, the frequency of technical errors, and the cognitive strain and workload faced by surgeons when using 2D systems. In addition, the price of a fully equipped 3D laparoscopic tower can be as little as a tenth of the usual cost of a surgical robot [8]. 

Traditionally, the major drawbacks of a laparoscopy have been considered to be rigid instruments with only 4 degrees of movement, which accentuates the fulcrum effect, and a steep learning curve [9]. Recent studies have proven that novice surgeons perform tasks with a superior outcome and time record when using 3D vision as compared with conventional laparoscopy [10]. In addition, 3D vision shortens the learning curve and adds precision when performing tasks previously considered difficult, such as suturing and needle guidance and transfer, making laparoscopy a viable option for routine procedures. The problems that arise are in the complex surgeries that involve extensive suturing and reconstruction [11]. Procedures such as a radical nephrectomy with thrombectomy, a nephroureterectomy with bladder cuff excision, and a radical cystectomy with intracorporeal urinary diversion are often considered off limits for the laparoscopic approach. 

The aim of the current study was to assess the feasibility of a 3D laparoscopic approach for the most complex urological procedures, as a viable alternative to robotic surgery. 

## 2. Materials and Methods

### 2.1. Patient Selection 

We included in the current study all patients who had undergone complex 3D laparoscopic procedures in our department since January 2017, including a radical nephrectomy (LRN) using the dual combined approach, a radical nephroureterectomy with bladder cuff excision (LRNU), and a radical cystectomy (LRC) with intracorporeal urinary diversion (ICUD). The study was conducted in accordance with the Declaration of Helsinki and all patients signed the informed consent form. All procedures were performed by a single surgeon (NC), who had 6 years of experience in open and robot-assisted uro-oncological surgery prior to the introduction of 3D laparoscopy in our department in 2017. Since then, all surgeries, except very few selected cases, have been performed by the laparoscopic approach. 

### 2.2. Surgical Technique

#### 2.2.1. Dual Combined Approach for Renal Tumors with Invasion of the Renal Vein or Inferior Vena Cava (IVC)

The dual combined laparoscopic approach comprised retroperitoneal access for artery and lumbar vein ligation, followed by transperitoneal access for completion of surgery and thrombectomy. For the retroperitoneal approach, three working trocars were positioned in a triangular manner as previously described [12,13] (Figure 1). An additional 5 mm additional trocar was placed for the assistant surgeon. 

After opening Gerota’s fascia, the renal artery was isolated, clipped, and sectioned. All the identified lumbar veins were sealed and divided and the posterior wall of the inferior vena cava (IVC) was extensively dissected. For the transperitoneal approach, three out of four trocars were repositioned (Figure 1). In some cases, a second 5 mm trocar was positioned in the epigastric area to facilitate liver retraction. The ipsilateral colon was dissected medially. The renal vein was isolated carefully in order to prevent tumor embolization. Dissection continued until the level of junction between the suprahepatic veins and IVC. On the right side, the central adrenal vein was clipped and divided. 

For level 0 thrombus patients, the tumor was milked entirely inside the renal vein, the latter being clipped as close to the IVC as possible, avoiding the cavotomy. For more advanced cases, the venous system was occluded using vessel loops and Rummel tourniquets in the following order: IVC below the renal pedicle, contralateral renal vein, and IVC above the thrombus. Cavotomy was performed, removing the entire tumoral thrombus. After heparinization, double-layered cavorrhaphy was performed, and the tourniquets undone in reverse order. The kidney was removed in an endobag through an inguinal incision. 

#### 2.2.2. Nephroureterectomy with Bladder Cuff Excision

The approach was transperitoneal and the trocars were placed in a standard manner (Figure 2). The trocar for the assisting surgeon was placed either in the epigastrium (right side) or midway between the costal rim and the anterosuperior iliac spine (left side). 

The ureter was clipped below the tumor to decrease the risk of caudal tumor seeding and a radical nephrectomy was performed, without kidney dissection from the lateral abdominal wall. 

In order to manage the distal ureter and to perform the excision of the bladder cuff, the surgical table was tilted to a Trendelenburg position and rotated to ensure 60° lateral decubitus. In some cases, an additional 5 mm trocar was inserted in the suprapubic area. For the pelvic dissection, the surgeon used the new additional trocar and the one situated above the iliac spine, whereas the paraumbilical trocar and the one located below the ribs were used for the camera. If no supplementary trocar was employed, the camera was repositioned through the trocar below the ribs and the main surgeon used as working trocars the one above the iliac spine or the former camera trocar. The isolation of the ureter was further extended, until its opening into the bladder wall was reached. Bladder cuff excision was performed after careful inspection of the inner wall of the bladder to identify the contralateral ureteral orifice. Before finalizing the partial cystectomy, the first suture was placed in order to prevent bladder retraction and ease further identification. The cystorrhaphy was carried out in a double-layered manner using a barbed suture, then verified by instilling 200–300 mL of saline.

#### 2.2.3. Radical Cystectomy with Intracorporeal Urinary Diversion

The patient was placed in dorsal decubitus, with the operating table tilted in the Trendelenburg position. Five trocars were used (Figure 3). 

A frozen section of the ureteral stumps was performed in all patients. The prostatectomy was carried out in a nerve-sparing fashion, when safe from an oncological standpoint. Lymphadenectomy was performed in all patients and comprised the removal of the obturator, internal, external and common iliac nodes. In female patients, the anterior vaginal wall was resected en bloc with the cystectomy specimen. 

For the urinary diversion, the left ureter was transposed on the right side through the mesosigmoid. Intracorporeal neobladder (NB) was constructed using modified Studer technique. The anastomosis with the urethral stump was done in the van Velthoven manner using barbed suture. To prevent tension in the anastomosis, the Rocco stitch was performed in some cases. Gastrointestinal continuity was reestablished using laparoscopic staplers. The ureters were anastomosed in Wallace 1 or 2 manner then catheterized using percutaneously inserted mono-J stents. The ureteral stents were inserted in the neobladder either through the anterior wall or through the suture. The watertightness of the reservoir was checked using 50 mL of sterile saline. 

For ileal conduit (IC) a 20 cm bowel segment was used. The ureters were sutured to the proximal end of the IC in Wallace 1 or 2 type anastomosis. Mono-J or double-J stents were used for ureteral catheterization.

### 2.3. Follow-Up

Follow-up after LRN with venous thrombus comprised chest and abdomen computed tomography (CT) scan at 3 and 6 months after surgery, followed by repeat imaging at every 6 months thereafter. Patients who underwent LRNU were followed-up by cystoscopy at every 3 months during the first two years and at 6 months thereafter. A chest and abdomen CT scan was performed every 6 months. In patients who underwent LRC, follow-up included a visit at 3–4 weeks after surgery for ureteral stent and/or Foley catheter removal. Imaging comprising chest and abdomen CT scan was performed at 3 and 6 months after surgery and at 6 months thereafter.

## 3. Results

A total of 19 patients (14 males, 5 women) underwent a combined approach LRN for renal cell carcinoma with venous thrombus. Their mean age was 67 years (range 53–80). In 73.7% of cases the tumor was located in the right kidney, and in 26.3% in the left. The mean tumor size was 7.9 cm (range 4.5–13). Two patients had a history of prior abdominal surgeries. A cavotomy was performed in 8 patients, all of whom had a right kidney tumor. In the other 11 patients, thrombus retraction after arterial clamping and retrograde milking allowed ligation of the renal vein, without the need for cavotomy. Perioperative data are summarized in Table 1. Postoperative staging was pT3a in 11 patients and pT3b in 8 patients. The rate of positive surgical margins (PSM) was 15.78% (3 cases), all due to thrombus microscopic invasion of the wall of the renal vein. The postoperative complication rate was 26% (5.26% Clavien ≥ III). Nine patients received adjuvant Sunitinib (3 with PSM and 6 with oligometastatic disease). Median follow-up was 18 months (range 3–26). At the last follow-up, 18/19 patients were alive. One patient died at 2 years after surgery with local and metastatic recurrence of the disease. 

LRNU with complete laparoscopic bladder cuff excision was performed for high-risk upper tract urothelial carcinoma in 13 patients (6 males, 7 females) with a mean age of 64 years (range 55–85). Perioperative data are summarized in Table 1. The mean number of lymph nodes was 3.5 (range 2–8). Pathology confirmed the presence of urothelial carcinoma, staged as pT1 in 5 patients, pT2 in 4 patients, and pT3 in 3 patients. In one patient, the ureteral tumor was identified as breast carcinoma metastasis. The rate of PSM was 7.69% (at the bladder wall for the patient with metastatic breast carcinoma). The mean catheterization time was 14 days (range 7–24). The postoperative complication rate was 30.76% (7.69% Clavien ≥ III). The median follow-up was 9 months (range 3–18). At the last follow-up, 11/13 patients were alive. During follow-up, one patient was diagnosed with chylous ascites for which repeated paracentesis was performed.

Twenty-one patients (19 males, 2 females) underwent LRC with ICUD. The mean age of the patients was 64 years (range 48–75). Intracorporeal NB was performed in 7 cases and IC in 14. Uretero-ileal anastomosis was performed in the Wallace 1 manner in 10 cases (4 IC, 6 NB) and the Wallace 2 manner in 11 cases (10 IC, 1 NB). Mono-J stents were used for ureteral drainage in all cases, except for three patients with IC, and were removed at 20 days (range 14–31) after surgery for NB and at 35 days (range 14–55) after surgery for IC. In patients with NB, the Foley catheter was removed at 28 days (range 21–30), following a control cystogram. The rate of 30-days postoperative complications was 23.8% (4.76% Clavien ≥ III). Eight patients were readmitted at a mean of 15.5 days for sepsis. Pathologic staging confirmed the presence of urothelial carcinoma in 19 cases and squamous cells carcinoma in 2 cases. The bladder tumor was staged as pT0 in 3 patients, pTis in 2 patients, pT1 in 5 patients, pT2 in 3 patients, and pT3 in 8 patients. The mean number of retrieved lymph nodes was 13 (range 6–33). Positive lymph nodes were encountered in 3 patients. One patient had a PSM (right lateral wall of the bladder, pT3b). The mean follow-up was 12 months (range 2–20). At the last follow-up, one patient had died due to stroke. All but one of the patients with NB were continent during the day at 6 months (using 0–1 pads).

## 4. Discussion

In the era of robotic surgery, interest in laparoscopy has decreased due to the steep learning curve and the work ergonomy for the surgeon. Despite its unquestionable advantages, however, robotic surgery takes an important toll on healthcare systems due to its significant cost. It is important not to deprive patients of the advantages of a minimally invasive approach. The data presented here support the feasibility of the 3D laparoscopic approach for the most complex uro-oncological procedures.

Radical nephrectomy with thrombectomy is considered by most practitioners as “the touchstone of urology”. The dual approach presented in this report delivers the benefits of each technique: the retroperitoneal access allows fast ligation of the renal artery, facilitating tumor ischemia and thrombus retraction and thus reducing the risk of embolization. On the other hand, the transperitoneal route ensures quick access to the venous system with minimal manipulation, while providing a generous working space to isolate the IVC and perform the thrombectomy. 

The dual combined laparoscopic approach has been shown to be feasible by other authors as well. Sanli et al. [14] reported the first two cases of use of the combined laparoscopic approach for radical nephrectomy and IVC thrombectomy in 2013. Using 3 trocars for each approach, they performed the intervention on tumors with level I thrombus extension. The total insufflation time was 150 and 165 min. In the first case, cavotomy and cavorrhaphy were performed, while in the second, milking the thrombus back into the renal vein and applying 2 Hem-o-Lok^TM^ clips sufficed. The patients were discharged at postoperative days 5 and 4. One of the patients developed lung metastasis 15 months after the surgery. Tang et al. [15] reported the case of a patient with a 6.4-cm right-sided tumor with level II thrombus who underwent a similar approach. No significant postoperative complications were reported by the authors. In contrast to previous reports, we employed 4 trocars for the retroperitoneal approach and 4 to 5 trocars for the transperitoneal approach. We believe that the additional 5-mm retroperitoneal trocar facilitated the exposure of the renal pedicle and IVC, while the transperitoneal ones provided multiple working channels for auxiliary instruments, reducing manipulation of the IVC and renal vein. 

Laparoscopic and robotic approaches have shown similar perioperative outcomes in patients with level I and II thrombus in terms of operative time, rate of major complications and oncologic outcomes (Table 2). However, there are still limits to use of the laparoscopic approach for IVC thrombectomy. Tumors with thrombus extending higher than the suprahepatic veins preclude the laparoscopic approach due to the impossibility of IVC isolation above the thrombus. In these cases, robotic surgery could represent a possibility. Chopra et al. [16] reported their experience in 11 patients with level III IVC thrombus, showing that the robotic approach enables isolation of the short hepatic veins and exposure of the retrohepatic IVC cranial to the thrombus. Furthermore, Gill et al. [17] reported their experience with level IV thrombectomy using the robotic approach. After securing the porta hepatis, infrarenal and retrohepatic IVC, and contralateral renal vein, a minithoracotomy was carried out in order to clamp the aorta and cardiopulmonary bypass was employed for complete thrombus extraction. Nonetheless, these procedures carry a significant risk of perioperative morbidity and thus should be reserved for highly experienced centers. 

When performing radical nephroureterectomy, it is crucial to remove the ureteral stump since the recurrence rate at this site, if not properly excised, ranges between 30% and 75% [25]. Even when the radical nephroureterectomy is performed by the minimally invasive approach, open access by Gibson incision is most frequently used for bladder cuff excision. However, the implantation of the ureter into the urinary bladder is deep in the pelvis and the open approach does not always confer the best vision. Consequently, the quality of the partial cystectomy remains questionable in some cases. Laparoscopy can ensure better access and visualization in the pelvis, allowing a completely endoscopic approach, as shown by our experience, with correct identification of all anatomic elements. Several reports have been published confirming the feasibility of a laparoscopic approach for nephroureterectomy with bladder cuff excision. Ghazi et al. [26] published a case series of 8 patients, using a supplementary trocar for pelvic access as well. In this series, the surgeon performed first the excision of the terminal ureter and bladder cuff, and then the radical nephrectomy. Indeed, having the kidney attached to the abdominal wall ensures tension on the ureter and facilitates its dissection. We prefer to perform the nephrectomy first, and leave the kidney attached only to the abdominal wall. After bladder cuff excision we immediately place the ureteral stump into the endobag, aiming to limit the contact between the mucosa and surrounding tissues in order to decrease the risk of peritoneal tumor seeding. 

Another possibility for performance of bladder cuff excision by the laparoscopic approach was described by Liu et al. [27] in a series of 31 patients. The authors reported use of a bulldog vascular clamp, which was modified to suit the curvature of the bladder, in order to clamp the perimeatic area. In this technique the forceps is inserted through a dedicated suprapubic trocar and approximates the walls, preventing urine spillage until the bladder cuff excision is performed. 

A completely retroperitoneal approach is also feasible for this procedure, as shown by Wu et al. in a study of 48 patients [28]. Four trocars were used for this technique, 3 in standard fashion plus an additional 5 mm trocar placed on the midclavicular line, 3 cm below the umbilical level. The dissection of the ureter in the pelvic segment and bladder cuff were managed using the additional port and the one placed above the iliac crest and was feasible in all cases. Moreover, Hattori et al. [29] suggest placing a stay suture at the anterior wall of the bladder and suspending it transabdominally. By applying tension to both the bladder and the ureter (which is grasped and tractioned cranially), the junction between the two structures is better exposed and circumferential dissection is facilitated in the retroperitoneal approach. 

As shown hereby, different techniques have been employed for nephroureterectomy with bladder cuff excision and have demonstrated the feasibility of a complete laparoscopic approach. When comparing series reporting laparoscopic and robotic radical nephroureterectomy with bladder cuff excision, no significant differences were observed for the operative time and major complications rate or oncologic outcomes (Table 3). On the other hand, the open approach for radical nephroureterectomy has shown lower 5-year cancer-specific survival as compared with minimally invasive techniques [30].

Primary T1 high-grade bladder tumors treated conservatively with transurethral resection and BCG protocols have a 64.9% rate of recurrence and 33% rate of progression during the first 36 months of follow-up [39]. Factors that can potentially influence these outcomes are obesity, tumor size greater than 3 cm, concomitant carcinoma in situ and increased preoperative neutrophil-to-lymphocytes ratio [40]. LRC with ICUD is a complex procedure [41]. Due to the limitations of laparoscopy in regard to reconstructive interventions, there is little experience in the field of ICUD, with few cases published in the literature [42,43,44,45,46,47,48,49,50,51] (Table 4). 

The ileal conduit is the most frequently performed diversion, while the ileal neobladder is mainly reserved for selected younger patients, with fewer comorbidities, who can withstand longer anesthesia and are able to follow post-operative recovery protocols. Even though minimally invasive surgery is challenging, it lowers the risk of significant post-operative complications, both local (wound seroma and dehiscence, abdominal eventration) and general (anemia, pain, ileus) [41]. 

In the present study, 7 out of 21 cases underwent LRC with intracorporeal neobladder. The challenge for this procedure is to perform the reconstruction of the neobladder by manual suture, taking into consideration surgeon tiredness after the excisional part of the surgery. Shao et al. [52] conducted a study on 55 patients, using a similar technique and number of trocars. For the initial cases, they performed manual suture of the ileal reservoir, eventually advancing to Stapler^TM^ systems. Despite simplification of the procedure by exclusion of the manual suture, the use of a stapler led to neobladder lithiasis in 2/30 patients. No significant difference was noted between the manually and the automatically sutured neobladder patients with respect to continence or urinary retention. 

Adamczyk et al. [53] analyzed 260 patients who underwent either laparoscopic or open radical cystectomy. The laparoscopic approach was superior in terms of median operating time and estimated blood loss. Postoperative complications had higher Clavien scores in the open group (laparoscopic versus open radical cystectomy: 2.24 versus 2.65, *p* = 0.0001), with 8 patients requiring reintervention for eventration, bleeding, urine leakage, infection, and peritonitis. However, when comparing a laparoscopy to robotics, there is no statistically significant difference in regard to oncological outcomes or major complications rate [54]. Also, similar functional outcomes have been reported (Table 4). 

The limitation of our study resides in the limited follow-up of the patients. Long-term oncological and functional outcomes are needed in order to validate any surgical approach. However, the aim of our preliminary study was to show the feasibility of a 3D laparoscopic approach for some of the most complex urological procedures.

## 5. Conclusions

In the era of robotic surgery, a laparoscopy remains a plausible alternative for most complex oncological cases. With increased experience, difficult interventions can be performed in a safe, cost-effective, and minimally invasive way. 

## Figures and Tables

**Figure 1 jcm-10-01812-f001:**
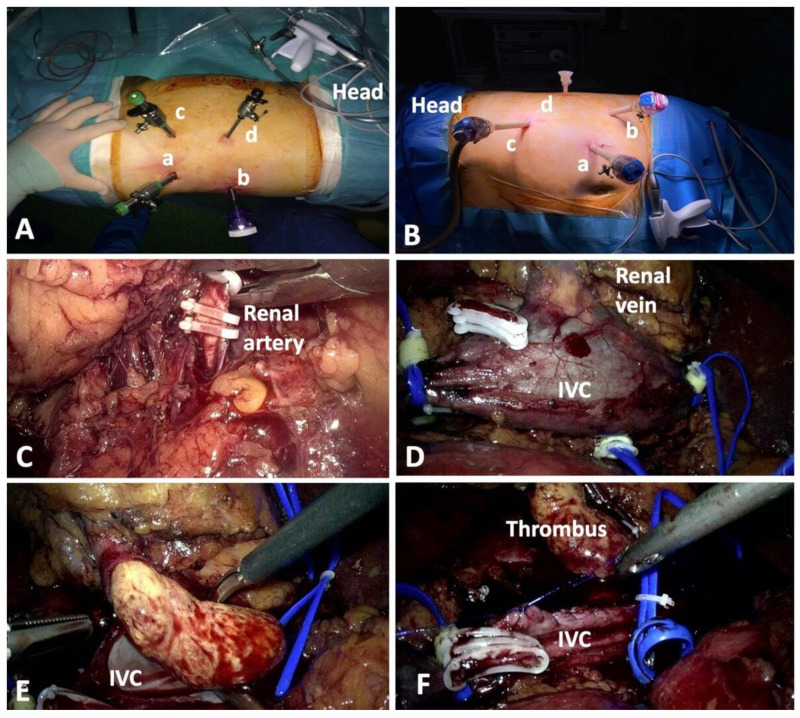
(**A**) Trocar positioning for the retroperitoneal approach in upper urinary tract surgery–left side. The 10 mm optic trocar (**a**) was placed on the mid-axillary line, above the iliac crest. Two 10 to 12 mm trocars were used, one located in the costomuscular angle (**b**), below the costal margin, and the second situated on the anterior axillary line, 2 cm above the anterior iliac spine (**c**). An additional 5 mm trocar was placed midway between the costal margin and the last trocar (**d**). (**B**) The optic trocar (**a**) was moved paraumbilically, at the lateral margin of the rectus abdominis muscle. One of the working trocars was placed on the line between the umbilicus and the anterior iliac spine (**b**), 2 cm cranially from the latter, and the second working trocar was inserted on the line linking the costal margin with the umbilicus at a right angle, 2 cm below the ribs (**c**). The additional 5-mm trocar used by the assisting surgeon was left in its initial location (**d**). (**C**) Clipping of the renal artery by retroperitoneal approach. (**D**) Inferior vena cava (IVC) isolation and occlusion. (**E**) Complete extraction of the tumor thrombus from the IVC by transperitoneal approach. (**F**) Final aspect of cavorraphy.

**Figure 2 jcm-10-01812-f002:**
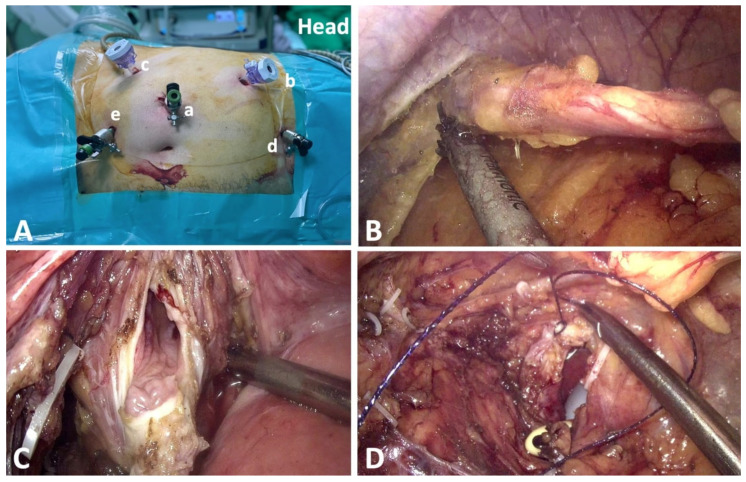
(**A**) Trocar positioning for right side laparoscopic transperitoneal radical nephroureterectomy with bladder cuff excision (**a**–**d**); suprapubic supplementary trocar used for pelvic dissection (**e**). (**B**) Isolation of the ureter in the pelvic segment. (**C**) Resection of the bladder cuff by the laparoscopic approach. (**D**) Cystorraphy using barbed suture.

**Figure 3 jcm-10-01812-f003:**
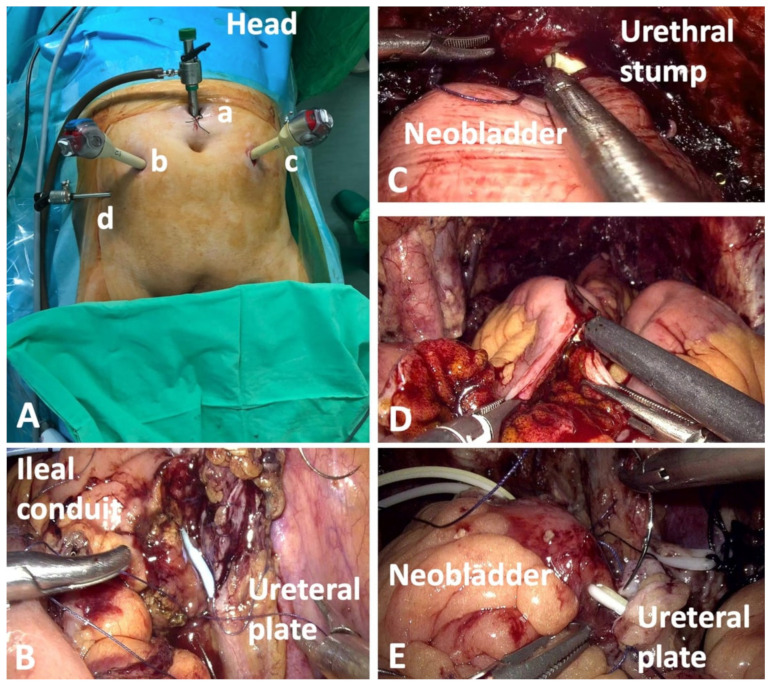
(**A**) Trocar positioning for laparoscopic radical cystectomy: the 10-mm optic trocar (**a**) was placed 2 cm superior to the umbilicus and two 12-mm instrument trocars were situated at the umbilical level, approximately 8 cm from the camera trocar, one on each side (**b**,**c**). An additional 5-mm trocar (**d**), for the assistant surgeon, was positioned above the iliac crest. (**B**) The anastomosis between the ureteral plate and the ileal conduit. (**C**) Anastomosis between the urethra and the neobladder performed by the laparoscopic approach. (**D**) Detubularization of the excluded intestinal loop. (**E**) The anastomosis between the ureteral plate and the neobladder.

**Table 1 jcm-10-01812-t001:** Three-dimensional complex laparoscopic procedures performed in our department.

No. of Patients	Preoperative Information	Procedure Specifics	Operative Time *, Minutes Mean (Range)	Blood Loss, mL Mean (Range)	Conversion, Reason	Length of Hospital Stay, Days Median (Range)	30 days Postoperative Complications	Oncologic Outcomes
Radical nephrectomy with thrombectomy by dual combined laparoscopic approach								
19	Mayo classification Level 0–8 patients Level I–7 patients Level II–4 patients	Cavotomy-8 patients No cavotomy–11 patients	Cavotomy: 361 min (280–555) No cavotomy: 330 min (175–400)	350 mL (150–600)	1 case, extension of the thrombus above the suprahepatic veins	6 (4–14)	Overall rate 26% Clavien I—3 patients (port site hematoma/prolonged lymphatic drainage) Clavien II—1 patient (transfusion) Clavien IV—1 patient (pulmonary thrombembolism)	PSM 15.78% New onset metastatic disease—2 patients Local and metastatic recurrence—1 patient (converted case)
Laparoscopic nephroureterectomy with bladder cuff excision								
13	4 patients with synchronous bladder tumor 3-TaG1 1-T1G3 (TURBT before surgery)	Transperitoneal approach	230.38 min (150–365)	100 mL (50–200 mL)	None	7 (4–8)	Overall rate 30.76% Clavien I—1 patient (surgical site infection) Clavien II—2 patients (transfusion) Clavien V—1 patient (death due to hemorrhagic diathesis)	PSM 7.69% Adjuvant chemotherapy—6 patients Bladder recurrence at 6 months—1 patient Metastatic progression of disease and death at 12 months (breast carcinoma)—1 patient
Laparoscopic radical cystectomy with intracorporeal urinary diversion								
21	MIBC-16 patients High-risk NMIBC-5 cases Neoadjuvant chemotherapy-10 cases No neoadjuvant chemotherapy in 6 cases of MIBC due to-histology (squamous cell carcinoma-2 cases), renal function impairment (4 cases)	NB–7 cases IC–14 cases	NB: 496.27 min (490–710) IC: 461.09 min (350–550)	200 mL (100–400 mL)	None	13 (6–29)	Overall rate 23.8% Clavien I—2 patients (delirium, urinary fistula in a patient with IC which was managed conservatively) Clavien II—2 patients (sepsis requiring antibiotic, digestive bleeding requiring transfusions) Clavien IIIb–one patient (reoperation for bleeding)	PSM 4.76% Local recurrence—1 patient New onset metastatic disease—1 patient

IC—ileal conduit, MIBC—muscle invasive bladder cancer, NB—neobladder, NMIBC—non-muscle-invasive bladder cancer, PSM—positive surgical margins, TURBT—transurethral resection of bladder tumor * Operative time included total operating room time (anesthesia and procedure).

**Table 2 jcm-10-01812-t002:** Literature overview of laparoscopic and robotic radical nephrectomy with IVC thrombectomy.

Author, Year	Number of Patients	Laparoscopic or Robotic Technique	Mayo Classification	Approach	Mean Operating Time (min)	Postoperative Complications	Oncologic Outcomes
Wang et al., 2014 [18]	5	Laparoscopic	Level I (*n* = 1) Level II (*n* = 4)	Retroperitoneal	127 (range 75–160)	None	PSM 0%
Shao et al., 2015 [19]	11	Laparoscopic	Level II (*n* = 6) Level IV (*n* = 5)	Level II: retroperitoneal Level IV: retroperitoneal and mini-thoracotomy	Level II: 155 (range 135–210) Level IV: 275 (range 260–310)	Overall rate 54.54% All Clavien I-II Level II (*n* = 2) Level IV (*n* = 4)	PSM–not reported New onset metastatic disease—1 patient
Tohi et al., 2019 [20]	5	Laparoscopic	Level I (*n* = 1) Level II (*n* = 3) Level III (*n* = 1)	Transperitoneal	316 (range 273–924)	Overall rate 20% Clavien III-V (*n* = 1)	PSM 0%
Cinar et al., 2019 [21]	13	Laparoscopic	Level I (*n* = 11) Level II (*n* = 2)	Transperitoneal	137.6 (range 60–200)	Overall rate 53.84% Clavien II (*n* = 4) Clavien IIIb (*n* = 1) Clavien IV (*n* = 2)	PSM 0%
Tian et al., 2020 [22]	78	Laparoscopic	Level 0 (*n* = 28) Level I (*n* = 27) Level II (*n* = 23)	Retroperitoneal Combined retro- and transpritoneal for left-sided level II thrombi Conversion rate 14.10% (*n* = 11)	256 (range 207–338)	Overall rate 16.66% Clavien I (*n* = 2) Clavien II (*n* = 9) Clavien IIIa (*n* = 2)	PSM 0% Local recurrence—1 patient New onset metastatic disease—8 patients
Andras et al., 2021 [Current paper]	19	3D Laparoscopic	Level 0 (*n* = 8) Level I (*n* = 7) Level II (*n* = 4)	Combined retro- and transperitoneal	330 (range 280–555) without cavotomy 361 (range 175–400) with cavotomy	Overall rate 26% Clavien I (*n* = 3) Clavien II (*n* = 1) Clavien IV (*n* = 1)	PSM 15.78% New onset metastatic disease—2 patients Local and metastatic recurrence—1 patient
Abaza, 2011 [23]	5 patients 6 thrombi	Robotic	Level I (*n* = 3) Level II (*n* = 3)	Transperitoneal	327 (range 240–411)	None	PSM 0%
Gill et al., 2015 [17]	16	Robotic da Vinci Si^TM^ and Xi^TM^	Level II (*n* = 7) Level III (*n* = 9)	Transperitoneal “IVC-first, kidney-last”	294 (range 270–378)	Overall rate 6.25% Clavien IIIb (*n* = 1)	PSM 0% New onset metastatic disease—2 patients
Chopra et al., 2017 [16]	24	Robotic da Vinci Si^TM^ and Xi^TM^	Level II (*n* = 13) Level III (*n* = 11)	Transperitoneal “IVC-first, kidney-last”	270 (range 180–480)	Overall rate 16.7% Clavien II (*n* = 2) Clavien IIIa (*n* = 1) Clavien IIIb (*n* = 1)	PSM 0% New onset metastatic disease—11 patients
Wu et al., 2021 [24]	35	Robotic da Vinci Si^TM^	Level I (*n* = 10) Level II (*n* = 15)	Retroperitoneal (*n* = 16) and transperitoneal (*n* = 19)	130 (range 100–250); 145 (range 110–275)	None	PSM 0%

**Table 3 jcm-10-01812-t003:** Literature overview of laparoscopic and robotic nephroureterectomy with bladder cuff excision.

Author, Year	Number of Patients	Laparoscopic or Robotic Technique	Approach	Mean Operating Time (min)	Postoperative Complications	Oncologic Outcomes
Ghazi et al., 2010 [26]	8	Laparoscopic Remote controlled robotic arm used for optic guidance	Transperitoneal “Ureter first, nephrectomy last” Extravesical bladder cuff	157 (range 110–200)	Overall rate 12.5% Bladder extravasation (*n* = 1)	PSM 8.33% Bladder recurrence—3 patients Local and metastatic recurrence—1 patient
Gillan et al., 2013 [31]	6	Laparoscopic	Transperitoneal Extravesical bladder cuff	190 (180–240)	None	PSM 0%
Liu et al., 2016 [27]	31	Laparoscopic	Transperitoneal Modified bladder cuff bulldog clamp	146.6 (range 90–257)	None	PSM 0% No recurrence
Wu et al., 2020 [28]	48	Laparoscopic	Retroperitoneal Extravesical bladder cuff	110 (range 100–130)	None	PSM 0% No recurrence
Miyake et al., 2020 [32]	4	Laparoscopic	Retroperitoneal Transvesical bladder cuff	174 (range 171–202)	Overall rate 50% Clavien I (*n* = 1) Clavien II (*n* = 1)	PSM 0% No recurrence
Ye et al., 2020 [33]	48	Laparoscopic	Transperitoneal with extravesical bladder cuff (*n* = 24) versus retroperitoneal with open bladder cuff (*n* = 24)	108.2 ± 11.2 versus 126.5 ± 10.8	Overall rate 8.33% Clavien I (*n* = 1) Clavien II (*n* = 3)	Not reported
Andras et al., 2021 [Current paper]	13	3D Laparoscopic	Transperitoneal Extravesical bladder cuff	230.38 (range 150–365)	Overall rate 30.76% Clavien I (*n* = 1) Clavien II (*n* = 2) Clavien V (*n* = 1)	PSM 7,69% Bladder recurrence—1 patient
Campi et al., 2011 [34]	66	Robotic da Vinci Si^TM^	Transperitoneal Extravesical robotic bladder cuff (*n* = 30) Open bladder cuff (*n* = 27) Transvesical bladder cuff (*n* = 5) Without bladder cuff (*n* = 4)	195 (range 180–270)	Overall rate 30.76% Clavien I (*n* = 16) Clavien II (*n* = 9) Clavien IIIa (*n* = 2) Clavien IIIb (*n* = 2)	PSM 6%—4 patients Bladder recurrence—16 patients New onset metastatic disease—5 patients
Hemal et al., 2011 [35]	15	Robotic da Vinci Si^TM^	Transperitoneal	184 (range 147–250)	None	PSM 0% No recurrence
Badani et al., 2014 [36]	26	Robotic	Transperitoneal	230 (range 120–310)	None	PSM 0% Bladder recurrence—4 patients
Zargar et al., 2014 [37]	31	Robotic	Transperitoneal	300 ± 69	Overall rate 19.35% Clavien I (*n* = 4) Clavien II (*n* = 1) Clavien III (*n* = 1)	PSM 3,2% Bladder recurrence—7 patients New onset metastatic disease—4 patients
Argun et al., 2016 [38]	2	Robotic da Vinci Xi^TM^	Transperitoneal	140; 150	None	PSM 0%

**Table 4 jcm-10-01812-t004:** Literature review of most recent series of laparoscopic and robotic radical cystectomy with complete intracorporeal urinary diversion.

Author, Year	Number of Patients	Laparoscopic or Robotic Technique	Type of Urinary Diversion	Mean Operating Time (min)	Postoperative Complications	Oncologic Outcomes	Functional Outcomes
Kanno et al., 2020 [42]	72	Laparoscopic	Ileal conduit	676 (range 612–740)	Overall first 30 days rate: 50% Clavien IIIa-V (*n* = 14) Overall 30–90 days rate: 28% Clavien IIIa-V: (*n* = 12)	PSM 1%–1 patient Local recurrence 8%–6 patients Distant recurrence 14%–10 patients Abdominal recurrence 6%–4 patients Upper tract recurrence 1%–1 patient	Not reported
Kubota et al., 2020 [43]	30	Laparoscopic	Ileal conduit	688 (range 641–740)	Overall rate: 13% Ileus (*n* = 1) Abdominal abcess (*n* = 4) Acute pyelonephritis (*n* = 2) Anastomotic leak (*n* = 1)	PSM 3%–1 patient	Not reported
Xu et al., 2021 [44]	12	Laparoscopic	Neobladder	414.6 ± 52.2	Overall rate: 41.66% Urinary tract infection (*n* = 1) Lymphorrhagia (*n* = 1) Ureteroenteric anastomotic stricture (*n* = 2) Metabolic abnormalities (*n* = 1)	PSM 0% No recurrence	Day-time continence at 12 months: 83.3% Night-time continence at 12 months: 58.3%
Andras et al., 2021 [Current paper]	21	3D Laparoscopic	Ileal conduit (*n* = 14) Neobladder (*n* = 7)	461.09 (range 350–550); 496.27 (range 490–710)	Overall rate: 23.8% Clavien I (*n* = 2) Clavien II (*n* = 2) Clavien IIIb (*n* =1)	PSM 4.76%–1 patient Local recurrence: 4.76%–1 patient New onset metastatic disease: 4.76%–1 patient	Neobladder: Day-time continence at 6 months: 85.71%—6 patients
Gok et al., 2019 [45]	98	Robotic	Neobladder	493.2 (range 258–750)	First 30 days–rate 51.02%: Clavien I-II (*n* = 30) Clavien IIIa-V (*n* = 20) 30–90 days–rate 13.26%: Clavien I-II (*n* = 6) Clavien IIIa-V (*n* = 7)	PSM 2%–2 patients Distant metastatic disease 14.2%–14 patients	Day-time continence: 60.6%—37 patients Night-time continence: 40.9%—25 patients Mean International Index of Erectile Function score in those with no previous erectile dysfunction: 20.6
Brassetti et al., 2019 [46]	137	Robotic	Neobladder	300 (range 240–350)	Clavien III-V (*n* = 15) Readmission in the following year: 18 (13%) patients	PSM 3%–4 patients Disease recurrence at 12 months: 7%	Day-time continence at 12 months: 79%
Tan et al., 2019 [47]	59	Robotic	Ileal conduit	330 (range 300–368)	Overall first 30 days rate: 48.4% Clavien IIIa-V (*n* = 5) Overall 30–90 days rate: 16.2% Clavien IIIa-V (*n* = 5)	PSM 8.5%–5 patients	Not reported
Hosseini et al., 2020 [48]	158	Robotic da Vinci Si^TM^	Neobladder	359 ± 98	Overall rate 23% First 30 days: Clavien IIIa-V (*n* = 29) 30–90 days: Clavien IIIa-V (*n* = 8)	PSM 1%–2 patients Tumor recurrence in 26 patients (41 sites)	Not reported
Porreca et al., 2020 [49]	83	Robotic	Ileal conduit (*n* = 32) Neobladder (*n* = 51)	410 (range 351–460)	Overall first 30 days rate: 35% Clavien I-II (*n* = 33) Clavien IIIa-V (*n* = 9) Overall 30–90 days rate: 20% Clavien I-II (*n* = 9) Clavien IIIa-V (*n* = 10)	PSM–3%–3 patients	Day-time continence at 12 months: 90.2%—46 patients Night-time continence at 12 months: 70.6%—36 patients Potency rate at 12 months: 31%—31 patients
Cacciamani et al., 2020 [50]	270	Robotic	Ileal conduit (*n* = 177) Neobladder (*n* = 93)	432.5 (range 379.7–489.2)	Overall first 30 days rate: 59.6% Clavien I-II (*n* = 119) Clavien IIIa-V (*n* = 42) Overall 30–90 days rate: 33.7% Clavien I-II (*n* = 62) Clavien IIIa-V (*n* = 29)	PSM 4.4%–12 patients Overall recurrence rate at 12 months: 50 (18.5%) patients	Not reported
Dell’Oglio et al., 2021 [51]	164	Robotic	Ileal conduit (*n* = 146) Neobladder (*n* = 18)	350 (range 327–360)	Overall rate: 35% Clavien II-V (*n* = 57)	PSM 7%-11 patients Recurrence at 18 months: 33 (20%) patients	Neobladder: Day-time continence: 78%—14 patients Night-time continence: 50%—9 patients

## Data Availability

All the data supporting the result are available upon request from the corresponding author.

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
