# Peer review of "Role of the Laparoscopic Approach for Complex Urologic Surgery in the Era of Robotics"

_jcm, 2021, doi:10.3390/jcm10091812_

Round 1
Reviewer 1 Report
Dear authors - congratulations on your work!
I think your work is a great summary of approaches to complex oncologic procedures with a pure laparoscopic approach - rarely done these days and requiring a significant amount of skill and experience.
I suggest you would look to compare your claims of ability to perform these procedures by comparing outcomes/approaches to traditional 2D laparoscopy and robotics in more detail. Nobody questions the possibility of complex surgery laparoscopically but the time it takes and complications is the concern. I feel the positive margin rate is a bit high, but this might comparable for robotic as well - this should be discussed in more detail. In general some comparison should be made to robotic series of the same complex oncologic procedures, as well as operative times. A table comparing their 3D laparoscopy with other types (2D and robotic) for the same procedures might be interesting.
Otherwise I think this is well written and more pictures would even help for a step-by-step of each procedure! Overall great job!
Author Response
Point 1: I suggest you would look to compare your claims of ability to perform these procedures by comparing outcomes/approaches to traditional 2D laparoscopy and robotics in more detail. Nobody questions the possibility of complex surgery laparoscopically but the time it takes and complications is the concern. I feel the positive margin rate is a bit high, but this might comparable for robotic as well - this should be discussed in more detail. In general some comparison should be made to robotic series of the same complex oncologic procedures, as well as operative times. A table comparing their 3D laparoscopy with other types (2D and robotic) for the same procedures might be interesting.
Response 1: We thank the reviewer for the kind words. Indeed, a comparison with robotic approach in terms of perioperative outcomes can support the use of laparoscopy for such complex cases. We added three tables (2,3,4) to illustrate the comparison between laparoscopy and robotics for these complex cases. Similar outcomes were observed in terms of operative time, complications rate and oncologic follow-up. Also, we added this supplementary information in the Discussions section – page 9, lines 608-610; page 11, lines 669-671; page 16, lines 809-811, Tables 2-4.
Point 2: Otherwise I think this is well written and more pictures would even help for a step-by-step of each procedure! Overall great job!
Response 2: We thank the reviewer for this suggestion. We added several intraoperative images – please see Fig 1-3.
Reviewer 2 Report
The authors present their experience with 3D laparoscopic surgery for Advanced urological oncology procedures (renal thrombus surgery, Nephroureterectomy, Radical Cystectomy.
-The manuscript provides a great deal of technical detail but it seems long to read and it should be shortened.
-The authors should define their previous surgical experience in urological oncology surgery before 2017, in order to clearly state that 3D could be an option over robotics for other groups.
-Complications should be presented as per Clavien classification.
-Follow for this patient shouls be described, as well as, the actual oncologic outcome. In the cystectomy group, threader needs to know about indication of neoadyuvant chemotherapy of not. In the renalal thrombus group the reader needs to know about any further treatment indicated. This would be the same for the nephroureterectomy group.
Author Response
Point 1: The manuscript provides a great deal of technical detail but it seems long to read and it should be shortened.
Response 1: We thank the reviewer for his/her comments. An important, but significantly long part of our article is the description of the surgical technique, taking into consideration that we aimed to present the three advanced laparoscopic procedures is such detail as to be reproducible. However, we agree that a too long paper might determine the reader to loose interest. Thus, we reduced the text that described standard approach – please see Surgical technique chapter.
Point 2: The authors should define their previous surgical experience in urological oncology surgery before 2017, in order to clearly state that 3D could be an option over robotics for other groups.
Response 2: We thank the reviewer for her/his comments. A single surgeon performed all these procedures. Before starting 3D laparoscopic approach in 2017, he had 6 years of experience in open and robot-assisted uro-oncological surgery. We added this information in Materials and Methods – page 2, lines 74-78.
Point 3: Complications should be presented as per Clavien classification.
Response 3: We thank the reviewer for his/her comments. All complications encountered in our cohort are presented as per Clavien classification in Table 1 (number of patients, Clavien classification and specific complication). We also added supplementary information regarding the rate of major complications (Clavien III or higher) in the Results section – page 6, line 514; page 6, line 526; page 6, line 536.
Point 4: Follow for this patient shouls be described, as well as, the actual oncologic outcome. In the cystectomy group, threader needs to know about indication of neoadyuvant chemotherapy of not. In the renalal thrombus group the reader needs to know about any further treatment indicated. This would be the same for the nephroureterectomy group.
Response 4: We thank the reviewer for her/his comments. Follow-up after LRN with venous thrombus comprised chest and abdomen CT scan at 3 and 6 months after surgery, followed by repeat imaging at every 6 months thereafter. Patients who underwent LRNU were followed-up by cystoscopy at every 3 months during the first two years and at 6 months thereafter. Chest and abdomen CT scan was performed at every 6 months. In patients who underwent LRC follow-up included a visit at 3-4 weeks after surgery for ureteral stent and/or Foley catheter removal. Imaging comprising chest and abdomen CT scan was performed at 3 and 6 months after surgery and at 6 months thereafter. We detailed this exact schedule in Materials and methods – page 5, lines 365-502.
In the cystectomy group, 10 of the 16 patients with MIBC underwent neo-adjuvant chemotherapy, this information being mentioned in Table 1. Six patients with MIBC did not undergo neoadjuvant chemotherapy due to: squamous cell carcinoma (2 cases) and renal function impairment (4 cases). We added this information in Table 1.
Of the 19 patients with renal tumor and venous thrombus, 9 underwent adjuvant targeted therapy with Sunitinib (3 patients with positive surgical margins and 6 with oligometastatic disease). We added this information in the Results section – page, line. Further oncologic outcomes are mentioned in Table 1.
Of the patients who underwent radical nephroureterectomy, six received adjuvant chemotherapy for muscle-invasive disease. Recurrence rate and metastatic progression are detailed in Table 1.
Round 2
Reviewer 2 Report
No further comments